# BaDLoss: Backdoor Detection via Loss Dynamics

## Abstract

Backdoor attacks often inject synthetic features into a training dataset. Images classified with these synthetic features often demonstrate starkly different training dynamics when compared to natural images. Previous work has identified this phenomenon, claiming that backdoors are outliers (Hayase et al., 2021) or particularly strong features (Khaddaj et al., 2023), consequently being harder or easier to learn compared to regular examples. We instead identify backdoors as having *different*, anomalous training dynamics. With this insight, we present BaDLoss, a robust backdoor detection method. BaDLoss injects specially chosen probes that model anomalous training dynamics and tracks the loss trajectory for each example in the dataset, enabling the identification of unknown backdoors in the training set. Our method effectively transfers zero-shot to novel backdoor attacks without prior knowledge. Additionally, BaDLoss can detect multiple concurrent attacks, setting it apart from most existing approaches. By removing identified examples and retraining, BaDLoss eliminates the model's vulnerability to most attacks, far more effectively than previous defenses.

## 1 Introduction

Current deep learning models rely heavily on large-scale datasets, often obtained through web scraping with minimal curation. These datasets are vulnerable to attackers, who can easily inject data and consequently alter the behavior of models trained on these datasets. Carlini et al. (2023) demonstrated that poisoning real-world, large-scale datasets is a feasible threat due to their distributed nature.

Among the various data poisoning threats, the creation of model backdoors is particularly insidious. By modifying only a small number of examples in a dataset, adversaries can make a trained model sensitive to highly specific features. The adversary can then control the model's outputs by injecting these features into otherwise innocuous images (Gu et al., 2017; Liu et al., 2018b; Barni et al., 2019; Li et al., 2021a; Carlini et al., 2023) – despite the model appearing benign during regular evaluation.

As attackers only inject a relatively small number of examples to evade detection, attackers aim to construct backdoors that are more easily learned than regular examples present in the training set. Nonetheless, few methods analyze the losses of backdoored examples. Notably, Anti-Backdoor Learning assumes that backdoored examples achieve a lower loss more quickly than normal examples (Li et al., 2021a). More recently, Khaddaj et al. (2023) instead claim that backdoor images have the *strongest* features, and are thus learned much more easily. Other works (Hayase et al., 2021) have looked at backdoors as anomalous examples which conflict with natural image features. While such claims are correct to an extent, we show that they can be refined. Backdoor images can exhibit both faster and slower training dynamics compared to a dataset's natural images, and we demonstrate that a more comprehensive treatment of loss dynamics can substantially improve backdoor defenses.

In this paper, we introduce ***BaDLoss***, a novel method that leverages differences in training dynamics to detect backdoor examples within the training set. Our approach is based on the observation that training dynamics vary significantly between clean and backdoor examples, as visually demonstrated in Fig. 3.

To generalize beyond assumptions that backdoors are simply easier or harder to learn, BaDLoss injects a small number of reference backdoor examples into the training set. A model is subsequently

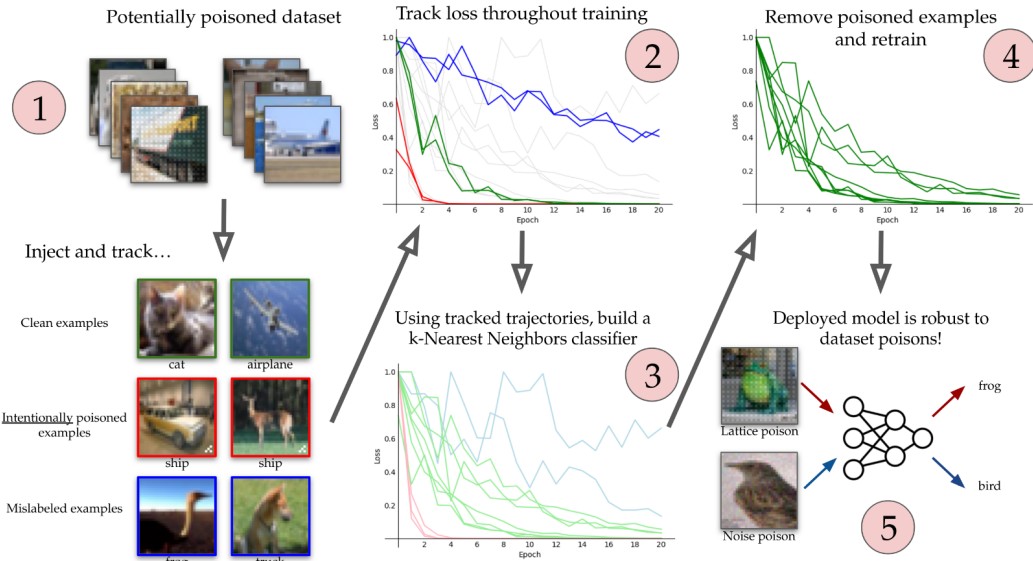

Figure 1: **BaDLoss Overview.** (1) The defender injects clean, poisoned, and mislabeled examples into the training set. (2) The defender tracks the loss on all examples as a model is trained on the *modified* training set. (3) The defender trains a k-NN with injected examples, classifying every training example as clean or anomalous. (4) The defender retrains the model, excluding any examples identified as anomalous. (5) The defender deploys the more robust model.

trained using all available examples, including the reference backdoors, while simultaneously tracking the loss associated with each instance in the training set. We use the injected backdoor examples and bonafide clean examples to define a k-nearest neighbors (k-NN) classifier on per-example loss trajectories. This classifier can then be used to identify unknown backdoor images within the training set. By carefully choosing the reference backdoor examples, we can directly model the anomalous loss dynamics we expect to see – whether easier or harder to learn than the natural examples in the training set. Unlike other defenses, we can extend to detect arbitrary attack types by adjusting the training dynamics induced by the reference backdoors.

BaDLoss demonstrates zero-shot transfer capability, which enables it to detect novel backdoor attacks. We hypothesize that, in order for attackers to effectively control a model, a backdoor must induce the model to attend to only a single feature for classification, unlike natural images. This unusual behavior induces differences in loss trajectories. Once backdoor examples are detected, we intervene by removing these instances from the training set and retraining the model. Many other defense methods (Li et al., 2021a; Chen et al., 2022) similarly detect backdoor examples before performing a more complex procedure on the model to remove the backdoor. As our detection is highly accurate and very general, BaDLoss is compatible with such methods. We demonstrate BaDLoss's effectiveness by comparing it to other backdoor defense methods, and highlight its superior performance.

In summary, the key contributions of our work are:

- Introducing a straightforward yet effective backdoor detection method by detecting anomalous training dynamics.

- Demonstrating that the method eliminates vulnerability against many attacks by identifying and removing backdoor examples before retraining.

- Showcasing zero-shot transfer to previously unseen backdoor attacks in a novel yet realistic multi-attack setting, which few existing defenses explicitly consider.

## 2 RELATED WORK

Since our work is concerned with backdoor detection and prevention, we will briefly discuss both attacks as well as defenses presented in the past.

### 2.1 BACKDOOR ATTACKS

Backdoor attacks in image classification can be broadly classified into two types: (1) attacks where the adversary provides a secretly backdoored model to the victim, and (2) attacks where the adversary modifies the training set used by the victim to train the model. We focus on the latter category, wherein the attacker manipulates a subset of the data so as to later control the model by injecting a feature of their choice.

Initial attacks modified both training images and their corresponding labels, such as adding a small stamp (Gu et al., 2017), overlaying another image (Chen et al., 2017), or incorporating a pattern (Liao et al., 2018) while changing the label of such images to the target class. Later attacks are more subtle, such as invisibly warping images (Nguyen & Tran, 2021), or overlaying image-specific invisible perturbations (Li et al., 2021d). In contrast to dirty-label attacks, clean-label attacks, initially relied on adding new training images with synthetic, simple-to-learn features that the learning algorithm would preferentially detect (Turner et al., 2019; Barni et al., 2019) without changing any image classes. More sophisticated strategies are harder to detect, such as using realistic reflection overlays (Liu et al., 2020), or random-noise patterns (Souri et al., 2022) to subtly induce the model to learn the desired behavior.

### 2.2 BACKDOOR DEFENSES

Various defense mechanisms have been proposed to mitigate model backdoors. One approach, known as trigger reverse engineering, assumes a low-magnitude trigger mask causes the model to misbehave. Neural Cleanse (Wang et al., 2019) and variants (Guo et al., 2019; Tao et al., 2022; Dong et al., 2021; Wang et al., 2020) extract the trigger, while others employ techniques like activation patching (Liu et al., 2019) or generative modeling (Qiao et al., 2019) to achieve similar effects.

Backdoor can be detected by directly examining model activations. Activation Clustering and Spectral Signatures (Tran et al., 2018) assume that the model activations separate clean from backdoored examples, and remove those examples before retraining. Further work examines specific properties of activation space and identifies influential neurons in backdoor attacks (Zheng et al., 2022; Wu & Wang, 2021; Xu et al., 2021). These neurons can then be pruned (Liu et al., 2018a), or the model fine-tuned appropriately (Chen et al., 2019; Li et al., 2021b; Zeng et al., 2022a).

As removing backdoors is challenging (Goel et al., 2022), many defenses are indirect, such as preventing the backdoor from being learned at all (Hong et al., 2020; Borgnia et al., 2021; Huang et al., 2022; Wang et al., 2022; Chen et al., 2022). Anti-Backdoor Learning (Li et al., 2021a) *maximizes* the loss of identified poisoned examples during training, making the model ignore poisoned features. Other approaches detect inputs that cause misbehavior in the output of the models (Gao et al., 2020; Chou et al., 2020; Kiourti et al., 2021) or exploit the unique frequency spectra of attack to identify attack inputs (Zeng et al., 2022b). Yet another line of work modifies inputs before the model sees them, reducing backdoor susceptibility (Doan et al., 2020; Do et al., 2023; Li et al., 2021c). Finally, some methods coarsely detect whether a network is backdoored at all (Xu et al., 2020; Kolouri et al., 2020).

## 3 METHODS

### 3.1 THREAT MODEL

In our threat model, we assume that the attacker has control only over the training dataset. They can view the entire dataset and modify a portion of it. This modified fraction is referred to as the poisoning ratio $p$. We assume that the attacker has no control over any other aspect of the training process. This corresponds well to an attack setting in which the victim trains a model using their own internal, well-tested code, but relies on a large, externally sourced dataset that cannot be manually

quality-checked. Conversely, we assume that the defender has complete control over the training process. However, they are provided with a labeled dataset from an unvalidated external source. These assumptions align with threat models considered in prior works (Carlini et al., 2023; Chen et al., 2018; Li et al., 2021a; Do et al., 2023).

Additionally, we assume the defender has access to a small set of guaranteed clean examples. Many other defense methods (Liu et al., 2018a; Wang et al., 2019; Gao et al., 2020; Chou et al., 2020; Kiourti et al., 2021; Li et al., 2021b; Zeng et al., 2022b; Wang et al., 2022; Huang et al., 2022) make the same assumption, as human labor could manually filter a subset of the dataset.

## 3.2 BaDLoss

Our approach draws inspiration from the Metadata Archaeology via Probe Dynamics (MAP-D) technique (Siddiqui et al., 2022), which examines per-example loss trajectories in a training set to remove unusual or mislabelled examples. Notably, other studies have also harnessed training dynamics for various objectives (Kaplun et al., 2022; Liu et al., 2022; Rabanser et al., 2022).

BaDLoss relies on the observation that backdoor triggers require the model to learn features absent in a typical dataset. These features are always aberrant, as the attacker must be able to control and inject the feature into an arbitrary image in order to successfully control the backdoored model after deployment. Consequently, such examples typically exhibit highly distinct loss dynamics, as illustrated in Fig. 3 in the Appendix. Our goal is to leverage loss trajectory analysis techniques to accurately identify and eliminate backdoored instances within the training set.

The BaDLoss algorithm is visualized in Fig. 1, expressed in five steps. A more rigorous breakdown can be seen in Section A.1 in the Appendix. The first step inserts probe examples into the training set. We select 500 bonafide clean examples and divide them into two sets: clean probes and backdoor probes. The 250 clean examples are left unchanged. For the 250 examples in the backdoor probe set, we change their labels and intentionally add a backdoor feature that we expect to show abnormally fast training dynamics. Additionally, we select 250 examples and randomize their labels, which we expect to show abnormally slow training dynamics (mislabeled examples). This allows us to capture backdoors which are learned either faster or slower than clean training examples.

The second step trains the target model, tracking the training dynamics of every example (including added probe examples) in the training set by recomputing the loss on each example after each epoch. With training dynamics for each example, the third step defines a k-nearest neighbors (k-NN) classifier to map loss-trajectories to a binary clean vs. backdoor classification, using the known labels of the added probe examples [1]. We assign clean probe examples to the clean class, and the backdoor probe class as well as the mislabeled probe class to the backdoor class for our k-NN classifier.

The fourth step uses the k-NN classifier to meta-label the entire training set. This allows us to filter the entire training set based on the probability of every example being a backdoored instance. Choosing a small threshold defines an aggressive data pruning strategy, which provides better protection while incurring a higher loss in the clean accuracy of the model. Choosing a lower threshold results in negligible loss in model performance, but may also fail to defend against strong adversaries where only a few examples are sufficient to successfully install the backdoor. The model is retrained on this filtered version of the dataset. Fifth and finally, this resultant clean model can be deployed, with reduced susceptibility to backdoor attacks from its training set.

## 3.3 Evaluation

To demonstrate BaDLoss's efficacy compared to previous work, we evaluate our method in two settings. In both settings, we evaluate standard metrics: (1) Clean accuracy of the retrained model, which indicates that the method does not degrade performance overmuch, and (2) Attack success rate against the retrained model, which ideally shows that the attack no longer functions against the retrained model.

**One-attack evaluation.** The majority of backdoor defense methods are evaluated against a single attack in their datasets (Wang et al., 2019; Chen et al., 2018; Tran et al., 2018; Li et al., 2021a). This

---

[1]We use simple $\ell_2$ distance, which disregards any fixed permutations of the sequence. Although other more sophisticated metrics can be explored, we found this simple approach to be sufficient for our purposes.

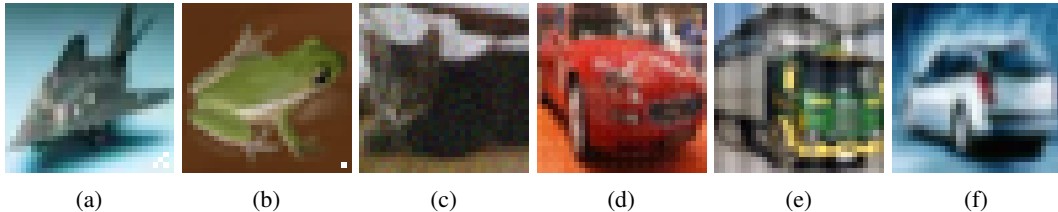

| (a) | (b) | (c) | (d) | (e) | (f) |

Figure 2: Different kinds of attacks considered in this work, including (a) checkboard pattern trigger (Patch) Gu et al. (2017), (b) single pixel trigger (Single-Pix) Gu et al. (2017), (c) blend attack with random noise (Blend-R) Chen et al. (2017), (d) blend attack with dimple pattern (Blend-P) (Liao et al., 2018), (e) blend attack with sinusoid pattern (Sinusoid) (Barni et al., 2019), and (f) warping field attack (Warping) (Nguyen & Tran, 2021).

is the canonical setting in the literature. In this setting, we set our injected backdoor examples to be similar to the chosen attack.

**Multi-attack evaluation.** The assumption that only one attack will exist in a dataset is unrealistic in most application areas, where multiple attackers could each be attempting multiple attacks to maximize their odds of gaining control over trained models. Furthermore, multiple attackers may be present, each using their own attack. Consequently, we test all methods in a scenario with simultaneous attacks. In our multi-attack evaluation setting, all attacks are present once in the training set. To simulate a lack of foreknowledge of how a dataset may be attacked, BaDLoss always uses an attack similar to the patch attack for its intentionally injected backdoor probe in the probationary run.

## 4 EXPERIMENTAL SETUP

We compare against a wide variety of classic and state-of-the-art backdoor attacks and defenses in the literature. Following recent work (Wang et al., 2022; Kiourti et al., 2021; Li et al., 2021b; Do et al., 2023), our evaluation focuses on the standard computer vision datasets CIFAR-10 (Krizhevsky, 2009) and GTSRB (Houben et al., 2013) which captures variance both in image size and class distribution.

### 4.1 ATTACKS CONSIDERED

In this study, we focus on attacks that exclusively impact the training dataset. While other attacks exist, our method is not designed to defend against such threat models. Fig 2 illustrates the attacks considered in this study.[2] Unless otherwise specified, all attacks are executed with a poisoning ratio of 1% of the total dataset size.

**Image Patch.** Gu et al. (2017) introduced one of the earliest and simplest backdoor attacks. In their work, they introduced a 4-pixel checkerboard pattern (Patch) and a single-pixel (Single-Pix) attack, wherein a fraction of the training images have the corresponding pixels changed to white, and their labels changed to the target class. In GTSRB, we use 2% for the checkerboard patch and 4% for the single-pixel patch instead of 1%.

**Blended Pattern.** In a blended pattern attack, a full-image trigger $t$ is blended into the image with some fraction $\alpha$, such that the attacked image $x_{\text{attacked}} = \alpha t + (1 - \alpha) x_{\text{original}}$. Chen et al. (2017) introduced this attack using a uniform random pattern (Blend-R), which we blend with $\alpha = 0.075$. We also apply alpha-blending to a fixed dimple pattern (one pixel affected every other row and column) (Liao et al., 2018) with $\alpha = 0.025$ (Blend-P). Additionally, we also blend in a sinusoid pattern (Barni et al., 2019) with $\alpha = 0.075$ (Sinusoid). The sinusoid attack is a clean-label baseline – unlike Blend-R and Blend-P, the target class is not changed for images with the sinusoid pattern. Therefore, the attack is only applied to images of the target class. As GTSRB has imbalanced classes, we require that the chosen target class for the sinusoid attack has at least 1,000 images, of which the sinusoid backdoor is applied to at least 500. Elsewhere, the sinusoid backdoor has the typical 1% ratio.

---

[2]It is worth noting that we failed to replicate the Sleeper Agent attack (Souri et al., 2022) in the course of our research.

**Warping Field.** Nguyen & Tran (2021) proposed the warping field attack (Warping), which generates a low-magnitude warping field to distort the original image in a way imperceptible to humans. Despite its imperceptibility, the model still learns to recognize the warping itself. In CIFAR-10, we follow the recommended poisoning ratio of 10% from Nguyen & Tran (2021), and in GTSRB, we instead use 20%. Additionally, we choose stronger warping field parameters than originally indicated so that the backdoors are readily learned: $s = 0.75, k = 6$ for CIFAR-10, $s = 1.0, k = 8$ for GTSRB.

## 4.2 DEFENSES CONSIDERED

All the defenses considered in this work implement some degree of filtration on the training dataset. This permits us to fairly compare our methods by evaluating the clean accuracy and attack success rates of retrained models after removing identified poisoned examples.

**Neural Cleanse.** Neural cleanse solves an inverse optimization problem to reconstruct the backdoor, then subsequently unlearns it with a variety of techniques (Wang et al., 2019). We use their proposed filtering technique: finding the backdoor, identifying triggered neurons, then filtering the training dataset by removing images with high trigger neuron activations.

**Activation Clustering.** Activation clustering clusters the model's last-layer activations for each class under the assumption that backdoored activations naturally cluster into two different regions i.e., one with the regular examples, and one with the backdoored examples. Chen et al. (2018) primarily propose retraining the model for every idenfified cluster (equates to twice the number of classes), which is computationally infeasible for large datasets. Instead, we use their alternate silhouette score identification method to remove poisoned data.

**Spectral Signatures.** Spectral signatures leverages the eigen-decomposition of the feature correlation matrix and removes a fixed fraction (15%) of datapoints that are deemed most anomalous (Tran et al., 2018). As the method always removes a fixed fraction of datapoints, this inevitably incurs a significant cost in terms of clean accuracy.

**Frequency Analysis.** Frequency analysis demonstrates that many backdoor attacks can be easily identified by distinct signatures in the frequency spectrum of backdoored images, as most attacks introduced high-frequency artifacts (Zeng et al., 2022b). For a direct comparison with BaDLoss, we train the frequency-domain anomaly detector using the same sample probe images that are used by BaDLoss (representing the defender's incomplete model over possible attacks) in both the single and multi-attack settings.

**Anti-Backdoor Learning.** Anti-backdoor learning assumes that backdoored examples can achieve lower loss values than typical training examples in the first few epochs of model training (Li et al., 2021a). Identified examples are removed from the training process, and at the end of training, they perform a short "unlearning" phase in which they *maximize* the loss on the identified backdoor examples to remove any lingering learned association with the backdoor features. For a direct comparison with BaDLoss, we instead use their method to identify backdoor examples and retrain the model from scratch without the identified backdoor examples. We also increase the number of samples identified in order to compensate for the lack of an unlearning phase, from 1% of examples to 15% (matching with Spectral Signatures).

## 5 RESULTS

We split the evaluations into two distinct categories (see Section 3.3 for a detailed description of the evaluations considered).

Clean accuracy is the retrained model's performance on the test set. Attack success rate (ASR) is evaluated on the full test set *excluding* the target class, with the backdoor injected into every example to verify that the attack successfully changes the model's predictions. In all evaluations, we use a ResNet-50 architecture and train with no augmentations. Full training details are available in Section A.4 in the Appendix.

We use an aggressive threshold for BaDLoss and discard any examples with a k-NN indicated backdoor probability higher than 0.1. This aggressive threshold harms the model's clean accuracy. However, a more principled tuning based on the ROC curve is also possible.

|  |  | Patch | Single-Pix | Blend-R | Blend-P | Sinusoid | Warping |
|---|---|---|---|---|---|---|---|
| **CIFAR-10** | Neural Cleanse | 0.82 | 0.77 | 0.94 | 0.89 | 0.48 | 0.75 |
| | Activation Clustering | 0.36 | 0.38 | 0.01 | 0.35 | 0.27 | 0.09 |
| | Spectral Signatures | 0.66 | 0.51 | 0.70 | 0.70 | 0.32 | 0.41 |
| | Frequency Analysis | 0.86 | 0.53 | **0.95** | **1.00** | 0.48 | 0.78 |
| | Anti-Backdoor Learning | 0.27 | 0.37 | 0.14 | 0.05 | 0.25 | 0.00 |
| | BaDLoss | **0.98** | **0.80** | 0.94 | 0.91 | **0.84** | **0.85** |
| **GTSRB** | Neural Cleanse | 0.94 | 0.86 | 0.97 | 0.80 | 0.50 | 0.75 |
| | Activation Clustering | 0.23 | 0.14 | 0.02 | 0.00 | 0.13 | 0.16 |
| | Spectral Signatures | 0.18 | 0.21 | 0.15 | 0.17 | 0.35 | 0.28 |
| | Frequency Analysis | 0.08 | 0.32 | **1.00** | **1.00** | 0.39 | **0.99** |
| | Anti-Backdoor Learning | 0.00 | 0.00 | 0.08 | 0.22 | 0.00 | 0.04 |
| | BaDLoss | **0.99** | **1.00** | **1.00** | 0.98 | **0.89** | **0.99** |

Table 1: **One-attack detection AUROC results on CIFAR-10 and GTSRB.** While BaDLoss does not always have the highest detection AUROC, it has robustly good performance on a variety of different attacks embedded in the training distribution, though neural cleanse and frequency analysis also perform well, albeit with noteworthy holes in their detection.

|  |  | Clean Accuracy | | | | | | Attack Success Rate | | | | | |
|---|---|---|---|---|---|---|---|---|---|---|---|---|---|
|  |  | Patch | Single-Pix | Blend-R | Blend-P | Sinusoid | Warping | Patch | Single-Pix | Blend-R | Blend-P | Sinusoid | Warping |
| **CIFAR-10** | No Defense | 84.09 | 83.49 | 83.68 | 83.61 | 83.33 | 82.19 | 93.56 | 89.38 | 99.90 | 100.00 | 53.62 | 89.89 |
| | Neural Cleanse | 82.73 | 82.96 | 83.60 | 83.61 | 83.63 | 82.16 | 1.38 | 0.99 | 6.51 | 1.72 | 28.76 | 87.56 |
| | Activation Clustering | 83.25 | 83.78 | 83.74 | 83.43 | 83.79 | 82.36 | 1.29 | 89.97 | 99.04 | 98.82 | 38.71 | 89.71 |
| | Spectral Signatures | 82.26 | 81.66 | 82.60 | 82.26 | 81.21 | 79.53 | 1.32 | 57.47 | 2.42 | 2.14 | 50.98 | 89.36 |
| | Frequency Analysis | 78.77 | 77.83 | 70.93 | 72.40 | 71.23 | 54.86 | 87.16 | 85.10 | 3.48 | 2.60 | 67.06 | 4.46 |
| | Anti-Backdoor Learning | 79.95 | 80.54 | 79.45 | 80.11 | 79.30 | 75.73 | 93.30 | 76.80 | 97.14 | 99.99 | 64.28 | 90.93 |
| | BaDLoss | 82.84 | 79.13 | 74.87 | 74.85 | 55.12 | 63.97 | 0.74 | 1.86 | 2.52 | 2.29 | 0.31 | 7.23 |
| **GTSRB** | No Defense | 97.34 | 96.03 | 97.88 | 97.63 | 97.81 | 89.81 | 81.64 | 0.91 | 98.58 | 100.00 | 58.10 | 54.45 |
| | Neural Cleanse | 96.78 | 97.71 | 97.91 | 97.77 | 98.81 | 91.41 | 0.16 | 0.32 | 97.99 | 0.00 | 58.10 | 51.61 |
| | Activation Clustering | 96.57 | 96.71 | 97.26 | 97.59 | 97.68 | 91.75 | 0.12 | 0.33 | 97.63 | 99.30 | 61.84 | 60.95 |
| | Spectral Signatures | 97.04 | 95.80 | 97.78 | 97.37 | 96.97 | 89.68 | 64.93 | 0.78 | 96.87 | 98.99 | 75.56 | 57.33 |
| | Frequency Analysis | 78.90 | 8.88 | 95.75 | 63.10 | 23.56 | 97.20 | 13.48 | 60.17 | 0.00 | 0.09 | 0.93 | 0.03 |
| | Anti-Backdoor Learning | 94.86 | 92.26 | 97.87 | 98.46 | 96.36 | 86.65 | 0.85 | 3.32 | 98.51 | 100.00 | 74.26 | 52.96 |
| | BaDLoss | 94.76 | 97.54 | 95.61 | 95.98 | 86.42 | 93.10 | 0.03 | 0.06 | 0.19 | 0.01 | 56.37 | 0.29 |

Table 2: **One-attack setting: Clean accuracy and attack success rate after retraining on CIFAR-10 and GTSRB.** BaDLoss is highly effective at removing detected backdoor instances, usually minimizing the backdoor's efficacy without degrading performance more than necessary.

## 5.1 ONE-ATTACK RESULTS

In the one-attack setting, we use $k = 20$ for our k-Nearest Neighbors classifier. For the patch and single-pixel attacks, the perturbation is placed in the opposite corner of the image. For the sinusoid and dimple blended attacks, the pattern is rotated 90°. For the random blended and warped attacks, new random patterns and warping fields are generated. Notably, as the warped attack has a substantially higher training fraction than all other attacks, BaDLoss uses three times as many probes in the one-attack setting to defend against the warped attack, so that probe training dynamics more closely imitate the attacked images. As the frequency analysis defense also explicitly models backdoored images, frequency analysis is trained on the same probe set as BaDLoss.

Detection results are presented in Table 1. We see that BaDLoss outperforms other defense methods in overall detection of backdoor attacks. While other methods exceed our performance on certain attacks (notably frequency analysis on blended pattern backdoors, which have a very distinct frequency spectrum), our across-the-board performance exceeds any other defense method. Many of these defenses get an AUROC score below 0.5 – this is a result of coercing these defense methods to act as detection methods by sweeping across their detection thresholds in order to generate a ROC curve. While these defenses are better than randomly removing examples at their *precise* detection thresholds,

| | | Clean Acc. | Attack Success Rate | | | | | | |
|---|---|---|---|---|---|---|---|---|---|
| | | | Patch | Single-Pix | Blend-R | Blend-P | Sinusoid | Warping | Avg. ASR |
| **CIFAR-10** | No Defense | 81.39 | 95.16 | 91.01 | 98.09 | 99.99 | 62.00 | 93.80 | 90.01 |
| | Neural Cleanse | 81.39 | 95.16 | 91.01 | 98.09 | 99.99 | 62.00 | 93.80 | 90.01 |
| | Activation Clustering | 81.26 | 94.02 | 24.04 | 7.76 | 2.70 | 59.59 | 92.57 | 46.78 |
| | Spectral Signatures | 79.46 | 2.30 | 1.91 | 3.78 | 98.56 | 68.49 | 88.49 | 43.92 |
| | Frequency Analysis | 71.02 | 6.64 | 79.66 | 18.83 | 16.14 | 83.26 | 77.82 | 47.06 |
| | Anti-Backdoor Learning | 78.28 | 95.31 | 66.69 | 97.76 | 99.99 | 96.60 | 94.49 | 91.81 |
| | BaDLoss | 60.26 | 1.78 | 4.57 | 10.12 | 6.29 | 17.63 | 67.10 | **17.91** |
| **GTSRB** | No Defense | 87.29 | 88.76 | 82.36 | 97.47 | 100.00 | 70.84 | 52.43 | 81.91 |
| | Neural Cleanse | 86.37 | 0.04 | 1.23 | 0.07 | 0.07 | 49.44 | 53.20 | 17.34 |
| | Activation Clustering | 85.43 | 0.50 | 1.37 | 0.16 | 98.45 | 35.74 | 46.28 | 30.42 |
| | Spectral Signatures | 81.82 | 0.34 | 2.03 | 29.15 | 95.83 | 47.15 | 46.52 | 36.84 |
| | Frequency Analysis | 4.58 | 67.16 | 19.05 | 99.98 | 0.21 | 0.63 | 3.51 | 31.76 |
| | Anti-Backdoor Learning | 74.43 | 1.24 | 3.30 | 0.33 | 0.26 | 57.28 | 44.38 | 17.80 |
| | BaDLoss | 80.23 | 0.10 | 0.29 | 0.03 | 0.04 | 55.91 | 21.26 | **12.94** |

Table 3: **Multi-attack setting: Clean accuracy and attack success rate after retraining on CIFAR-10 and GTSRB.** This table shows that the multi-attack setting is substantially harder than the single-attack setting. BaDLoss demonstrates the best overall defense in both settings, but suffers some clean accuracy degradation in CIFAR-10.

their performance rapidly degrades as the detection threshold varies. In contrast, BaDLoss's detection threshold can be easily adjusted by the practitioner without drastic changes in performance.

Defense results after removing detected examples and subsequent retraining are presented in Table 2. We see that across the slate of attacks, BaDLoss tends to match or outperform all other defense methods – only failing to defend against the sinusoid attack in GTSRB. In order to defend against attacks by detecting and removing backdoored training examples, an extremely high degree of reliability is often needed; against the sinusoid attack in GTSRB, BaDLoss successfully removed 94.8% (474/500) of the backdoored examples. Nonetheless, the backdoor was still learned.

## 5.2 MULTI-ATTACK RESULTS

In the multi-attack setting, all attacks are simultaneously carried out against the defended model (see Section 3.3 for a more detailed description). We use $k = 30$ for our k-NN in CIFAR-10, and $k = 10$ in GTSRB. This is a parameter tunable by the defender. We suggest tuning this parameter through leave-one-out cross validation on the set of probe examples to find the $k$ value that maximizes the AUC score. As previously mentioned, we use an aggressive detection threshold of 0.1. As a probe, we only use the flipped patch probe described above. However, we note that the defender can choose to use a more complex probe set if they so prefer – as demonstrated in the previous section, having a probe whose loss dynamics closely matches the attack a defender expects to see can dramatically improve detection performance. Frequency analysis uses the same probe to train its anomaly detector.

The results are summarized in Table 3. As we see, every defense method but BaDLoss is susceptible to at least one attack at near-maximum attack success rate. Even high detection accuracy is not always sufficient to remove backdoors that have particularly high poisoning ratios. The warped attack specifically is hard to defend against as a result. In GTSRB, BaDLoss detected and removed over 70% of the warped training examples, but the remaining poisoned images were still enough for the model to learn the backdoor, albeit to a lesser extent.

## 6 Discussion

### 6.1 Impact of the Poisoning Ratio

As MAP-D leverages the training dynamics of the model, the poisoning ratio of a particular attack is highly impactful. For instance, the Patch attack is not learned at all on GTSRB at a 1% poisoning ratio, but is learned adequately at 2%. These poisoning ratios substantially affect the loss dynamics of the corresponding examples. For instance, the warped attack, which is ordinarily substantially harder to learn than ordinary training examples at a low poisoning ratio, instead becomes easier to learn when the poisoning ratio is increased. Incorrectly chosen probe examples can fail to detect examples with the opposite style of training dynamics. Hence, we use an increased probe ratio during our one-class evaluations for the warped attack. Naturally, changes in dataset size and model capacity can also affect the loss dynamics. While our tests have indicated BaDLoss is robust in common settings, we anticipate this may change at extremes of the training variables.

### 6.2 Interactions in the Multi-attack setting

Using multiple simultaneous attacks makes loss trajectories less stable as the model switches between recognition through one mechanism to another (Lubana et al., 2023). As BaDLoss relies on loss trajectories, this instability impacts its performance to some extent. Despite the instabilities of individual loss trajectories, we find that our method still works well.

Interestingly, simultaneous attacks can interact positively and improve each others' performance. Notably, in GTSRB, the single-pixel attack fails to generalize at all in the one-attack setting (and every defense trivially defends against it). However, in the multi-attack setting, the attack generalizes. We hypothesize this is due to the simultaneous patch attack in the same corner of the image. While the patch and single-pixel attacks target different classes, the patch attack teaches the network to attend to the bottom-right corner, potentially helping learn the single-pixel attack. We strongly recommend future work in model backdoors to evaluate in the multi-attack setting.

### 6.3 Limitations

We highlight some of the major limitations of our work.

**Impact of example removal.** Our defense methodology removes examples from the training set. This is a common practice in past work (Tran et al., 2018), but removal can have outsized negative impacts on the long-tail of minority groups present in a dataset (Feldman & Zhang, 2020; Liu et al., 2021; Sanyal et al., 2022). Therefore, it is important to understand the impact of removal on the resulting model beyond just the average accuracy of the model in sensitive domains.

**Requirement of probe examples.** We inject examples in the training set with desired training dynamics. If the target training dynamics are fully unknown, we can track only clean loss trajectories and detect potential attack trajectories by measuring the average distance to the clean trajectories and performing anomaly detection.

**Counter-attacks.** Generally, our assumption that attacks demonstrate anomalous loss dynamics is more robust than other models of attacks implied by other defenses (for instance, neural cleanse's model that an attack uses a low-magnitude trigger). However, a knowledgable attacker can still exploit this model with access to the training dataset, e.g. by carefully selecting injected features, poisoning ratios, and so on to ensure that their attack demonstrates a natural loss trajectory. Additionally, trained attack images (such as the sleeper agent attack (Souri et al., 2022)) could add a regularization term in training to ensure that their losses mimic clean examples.

## 7 Conclusion

This paper presents BaDLoss, a novel backdoor detection technique leveraging the training dynamics of the model. We show that BaDLoss is simpler and much more capable of detecting backdoor attacks in the real world, especially in the realistic multi-attack setting, where most prior methods fail. We also discuss the impact of various training settings on the resulting loss dynamics, and hence on the performance of BaDLoss.

## 8 ACKNOWLEDGEMENTS

Hidden for double-blind review.

## 9 ETHICS STATEMENT

Machine learning models have a known weakness on minority classes. For image models particularly, minority classes in certain domains can correspond directly to real-world minority groups, and model inaccuracies can cause real-world harms for these groups. Any backdoor defense method that attempts to remove datapoints can exacerbate problems induced by imbalanced class distributions. Not only may the examples be labeled anomalous, any detection method without a 0% false positive rate will inevitably remove examples from minority classes. BaDLoss is no exception, and should be used with care on such applications.

## 10 REPRODUCIBILITY STATEMENT

We use publicly available datasets, and all our code is available on GitHub: [page redacted for double-blind review].

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

# A APPENDIX

## A.1 ALGORITHM

Notation: Dataset $D$, potentially backdoored. Subset $C \in D$ of $2n$ bona-fide clean examples. Learning algorithm that takes a dataset $D$ and produces a learned $M$ and a time series of losses $L$ for each data point in the given dataset: $\mathcal{A} : D \to M, L$. Parameters $k \in \mathbb{N}$, $t \in [0, 1]$. A reference backdoor function $B : D \to D$ that takes datapoints in $D$ and adds a backdoor to them, possibly changing their labels.

1. Select half of the clean examples ($n$ examples) in $C$ as $P_1$. These examples will stay clean.
2. Select the remaining $n$ clean examples in $C$ and use $B$ to add a backdoor to them, producing $P_2$
3. Select $n$ additional examples from the dataset $D$ and randomize their labels so that every example is mislabeled – call this set of points $P_3$.
4. Inject $P_1, P_2, P_3$ into $D$, overwriting their previous points. Call this dataset $D'$
5. Train a model $M'$ on $D'$ using $\mathcal{A}$, producing a series of training losses $L$. $L$ contains a loss trajectory per example, where each loss trajectory is a sequence of real numbers of length equal to the number of epochs of training.
6. Select the losses for $P_1$, $P_2$, and $P_3$, treating the losses as Euclidean vectors. Call these losses the probe set. $P_1$ is the clean probe set, $P_2$ and $P_3$ are the anomalous probe set.
7. For each training examples in $D' \notin P_1 \cup P_2 \cup P_3$, find the $k$ nearest neighbors in the probe set, using Euclidean distance. If more than $t$ of the neighbors are from the anomalous set (i.e. either $P_2$ or $P_3$), mark the example as "Reject".
8. Remove every training example marked "Reject" from $D$, producing $D^*$.
9. Retrain on $D^*$, producing cleaned model $M^*$.

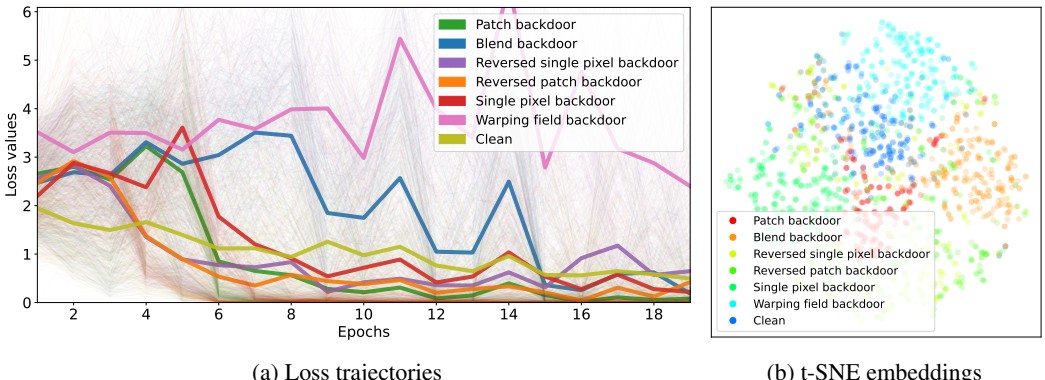

(a) Loss trajectories        (b) t-SNE embeddings

Figure 3: **Loss dynamics on CIFAR-10 with many simultaneous backdoor attacks** (Gu et al., 2017; Chen et al., 2017; Nguyen & Tran, 2021) (a) Visualization of the loss trajectories from the first 20 epochs. Fainted lines represent individual example trajectories while solid lines represent the mean trajectory. (b) t-SNE embeddings of the loss trajectories. The figure provides evidence that the training dynamics on backdoored examples are distinct from the clean examples without any backdoors.

## A.2 EXTENSION: TEST-TIME DETECTION

The algorithm in A.1 can be modified to permit test-time classification in the following way:

1. Using a test set of data $D_T$, create clean and poisoned probes similarly to on the training set.
2. When training $M'$, store checkpoints of the model at each epoch.
3. Once training has completed, save the loss-trajectories of the clean and poisoned *test* probes as a k-nearest neighbors classifier.
4. At test time, evaluate the test point on the sequence of checkpoints of $M'$.
5. Use kNN classification using the test probe loss trajectories to determine whether the example is backdoored or not.

## A.3 ADDITIONAL FIGURES

See: Figure 3

## A.4 TRAINING DETAILS

- Architecture: ResNet-50.
- Epochs: 100
- Batch Size: 128 (CIFAR-10), 256 (GTSRB)
- Optimizer: AdamW
- Learning rate: 1e-3
- Weight Decay: 1e-4
- LR Schedule: Cosine Annealment
- No augmentations

