# OpenReview forum: "BaDLoss: Backdoor Detection via Loss Dynamics"
_ICLR.cc/2024/Conference — Submitted to ICLR 2024_

### Official Review · Reviewer_QKR4 · 2023-10-21

**Soundness:** 3 good
**Presentation:** 2 fair
**Contribution:** 2 fair
**Rating:** 3
**Confidence:** 4

**Summary:**

This paper proposed a backdoor data detection method called BaDLoss, which is inspired by existing works on anti-backdoor learning (ABL) and Spectre. Instead of detecting backdoor data using lower per-example loss, BaDLoss treat the training trajectories over each epoch as a vector. BaDLoss select a subset of samples to inject a selected backdoor pattern as a reference. After training, it uses k-NN with Euclidean distance to filter out backdoored data. The proposed method demonstrated its effectiveness on existing attacks.

**Strengths:**

Using loss trajectories as a vector is new and interesting in this field. The motivation for using such an approach is well explained. It is technically sound for the proposed method.

**Weaknesses:**

My main concern on the weakness is the evaluations and practicality of the proposed method.
- The proposed method relies on predefined reference backdoor samples. This could limit its practicality.
- For one-attack evaluations, on page 4, section 3.3, reference examples are set to be similar to the chosen attack. This is impossible in a real-world scenario. The defender should not have any prior knowledge regarding backdoor attacks.
- It has been observed in existing works such as SPECTRE (Hayase et al., 2021) and [3] that the detection method is sensitive towards the poisoning rate. It would be more comprehensive to include experiments with lower poisoning rates. Some additional results to provide evidence for the discussion in section 6.1 would be great.
- It is unclear which model architecture is used in the evaluations and if the proposed method works on other models.
- Lack of comparison with more recent detection methods [1,2,3].
- In the introduction, page 2, below Figure 1. "This is because backdoor attacks generally cause the model to attend to a single feature for classification unlike natural images, which generally induces anomalous loss trajectories for those backdoor examples." This is an overclaim; there is no evidence to support this statement.
- The experiments were only conducted on the small-scale dataset, lacking evaluations on larger datasets and more recent attack ISSBA [4].
- Lack of evaluations against adaptive attacks. Given that the adversary knows that the defender will use BaDLoss, to what extent could the adversary evade detection? For example, if an adversary could have access to the entire training dataset and select several data points that have various loss curves (before adding backdoor triggers), would this evade detection?


[1] Chen, Weixin, Baoyuan Wu, and Haoqian Wang. "Effective backdoor defense by exploiting sensitivity of poisoned samples." Advances in Neural Information Processing Systems (2022).\
[2] Pan, Minzhou, et al. "ASSET: Robust Backdoor Data Detection Across a Multiplicity of Deep Learning Paradigms." USENIX Security Symposium (2023).\
[3] Huang, Hanxun, et al. "Distilling Cognitive Backdoor Patterns within an Image." The Eleventh International Conference on Learning Representations (2023).\
[4] Li, Yuezun, et al. "Invisible backdoor attack with sample-specific triggers." Proceedings of the IEEE/CVF international conference on computer vision. 2021.

**Questions:**

The results for ABL in Table 1 seem much lower than the results reported in their original paper, as well as in reproduced results in [3]. Is there any reason for this discrepancy?

---

> ### Author Response · Authors · 2023-11-20
> **Addressing Weaknesses**
>
> Thank you for the highly detailed feedback in your review. We appreciate your comments that our approach is novel, interesting, and well-motivated.
>
> > The proposed method relies on predefined reference backdoor samples. This could limit its practicality.
>
> In the multiattack setting, which we consider to be our most important evaluation, as it most closely mimics a practical setting where many attackers may deploy multiple attacks against a defender, we only use the simple Patch attack as our reference attack. Our robust results with only this reference show our strong generalization ability. This is because our probe set does not assume anything about the structure of the attack – solely that its learning dynamics will be different; that the attack will be either easier or harder to learn than normal training examples.
>
> If a defender has specific attack types they want to defend against, they could introduce those attacks into the probe-set to improve their robustness against that type of attack. We’ve added a note indicating this way in which the method could be extended.
>
>
> > For one-attack evaluations, on page 4, section 3.3, reference examples are set to be similar to the chosen attack. This is impossible in a real-world scenario. The defender should not have any prior knowledge regarding backdoor attacks.
>
> We consider this setting as evidence for the claim above; that a more informed defender will defend more effectively. You are correct that in general a defender will not know what attacks are deployed against them – we believe our strong performance in the multiattack setting demonstrates that BaDLoss works even when the defender is unaware of potential attacks.
>
>
> > It has been observed in existing works such as SPECTRE (Hayase et al., 2021) and [3] that the detection method is sensitive towards the poisoning rate. It would be more comprehensive to include experiments with lower poisoning rates. Some additional results to provide evidence for the discussion in section 6.1 would be great.
>
> We agree that it would be valuable to collect results that vary across poisoning ratios. However, we note most filtering-based defenses only affect a small band of poisoning ratios. In low poisoning ratios, even nearly-trivial filtering removes enough examples that the model memorizes backdoored train images and cannot generalize to new attack images at test time. In high poisoning ratios, filtering-based defenses that remove even 90% of poisoned examples can still fail to produce a clean model. Other defenses are needed in high poisoning ratios (possibly simply making the practitioner aware that they have a highly poisoned dataset, and that they should perhaps scrap the data altogether).
>
>
> > It is unclear which model architecture is used in the evaluations and if the proposed method works on other models.
>
> Apologies, and thank you for the catch. We’ve added a note specifying that we collected results primarily on ResNet-50s.
>
>
> > Lack of comparison with more recent detection methods [1,2,3].
>
> We agree that  broader baselines would improve the robustness of our results. However, we found that older, relatively simpler defenses (e.g. Neural Cleanse, Frequency Analysis) performed surprisingly well across a range of attacks. We believe that our set of defense baselines is strong evidence for the efficacy of our method.
>
>
> > In the introduction, page 2, below Figure 1. "This is because backdoor attacks generally cause the model to attend to a single feature for classification unlike natural images, which generally induces anomalous loss trajectories for those backdoor examples." This is an overclaim; there is no evidence to support this statement.
>
> Our apologies, and thank you for the feedback. We agree that this was an overclaim. We have adjusted this line to separate our conjecture from our experimental evidence.
>
>
> > The experiments were only conducted on the small-scale dataset, lacking evaluations on larger datasets and more recent attack ISSBA [4].
>
> Compute-availability limitations prevented us from performing evaluations on a maximally-broad set of datasets. We agree that more recent attacks could have been added, and that this would improve the robustness of our results.
>
>
> > Lack of evaluations against adaptive attacks. Given that the adversary knows that the defender will use BaDLoss, to what extent could the adversary evade detection? For example, if an adversary could have access to the entire training dataset and select several data points that have various loss curves (before adding backdoor triggers), would this evade detection?
>
> As we mention in the discussion section, an adversary can train against BaDLoss – e.g. by optimizing their triggers to ensure that backdoored images have normal loss dynamics. We have not evaluated whether an adversary can fool BaDLoss without direct optimization against it, perhaps with a method like you suggest. We leave such explorations to future work.

---

> ### Author Response · Authors · 2023-11-20
> **Answering Question Regarding Anti-Backdoor Learning**
>
> Finally, thank you for your question regarding Anti-Backdoor Learning. We agree that this is an important point, and would like to provide more context on the performance we report. We attribute the discrepancy to two reasons:
> 1. First, we note the algorithmic differences:
>     1. The method described in the paper (https://arxiv.org/pdf/2110.11571.pdf) is not the same as is provided in their codebase (https://github.com/bboylyg/ABL).
>         1. In the paper, the first phase identifies a subset of backdoored examples, and the second phase minimizes the loss on normal examples while maximizing the loss on the identified examples simultaneously.
>         2. In the codebase, the first phase is the same, but then training continues as normal for the second phase (without the gradient ascent objective!) and a third phase is added in which only gradient ascent is performed on the identified examples.
>     2. We were unable to replicate Anti-Backdoor Learning’s results using the method described in the paper. We were able to replicate their results using the method described in their codebase. However…
>     3. We use neither algorithm, as we use all defenses in the filter-and-retrain setting, to judge the quality of the filtering. Therefore, we used Anti-Backdoor Learning’s initial phase to identify backdoored examples, but then simply removed those examples and retrained the model from scratch.
>         1. This is because methods that identify backdoored examples then repair the poisoned model are inherently modular – one could easily apply the gradient ascent third phase of ABL to identified examples from Neural Cleanse or BaDLoss.
>         2. As a result, it is expected that Anti-Backdoor Learning’s performance would be worse in this setting than as reported in their paper.
> 2. Second, in our replication of Anti-Backdoor Learning’s codebase algorithm, we found that performance was dependent on particular hyperparameter settings.
>      1. As the goal is to compare how useful methods are to practitioners, we used out-of-the-box hyperparameters reported in their paper.
>      2. ABL’s performance could likely be improved by tuning its hyperparameters, possibly using attacks temporarily introduced by the defender to calibrate, but doing so is an involved and computationally expensive process, especially when the true distribution of attacks is unknown.

---

### Official Review · Reviewer_SuvN · 2023-10-30

**Soundness:** 2 fair
**Presentation:** 2 fair
**Contribution:** 2 fair
**Rating:** 3
**Confidence:** 3

**Summary:**

This paper presents BaDLoss, a new backdoor detection method that exploits the difference training dynamics between clean and backdoor samples by injecting specially chosen probes into the training data. These probes model anomalous training dynamics, and BaDLoss tracks the loss trajectory for each example in the dataset to identify unknown backdoors. By removing identified backdoor samples and retraining, BaDLoss can mitigate the backdoor attacks.

**Strengths:**

1. The proposed method works based on observing the significantly vary training dynamics between clean and backdoor samples, which is quite novel and interesting.
2. Overall, the method's performance seems to outperform other baselines.

**Weaknesses:**

1. [method's presentation, major] I personally find the presentation of Section 3 quite hard to follow since there are no algorithm or figure to describe the method, or even a formulation.
2. [lack of experiments, major] There are only 3 types of backdoor attack (patch-based, blending-based, warping-based) that are considered in the experiments, so I am not sure if the defense is effective with all attacks. I think there should be more backdoor attack approaches included in the experiments as well as related works discussion, such as sample-specific ([1]), optimized trigger ([2]), or frequency domain attack ([3]). Moreover, there is no abaltion study/discussion about different choices for the hyperparameters used in the paper (detection threshold, k for the kNN classifier).
3. [underwhelming experimental results, major] The clean acc. of BaDLoss is significantly degraded in the cases of SIG, WaNet, and multi-attack on CIFAR10. With those underwhelming clean acc. (~60%), I doubt that the model can be considered functional, especially on such "easy" dataset like CIFAR10.
4. [results' presentation, minor] There are many numbers in Table 2 and Table 3 but the best results are not highlighted. The authors should highlight the best results, or maybe report the average clean acc. and ASR drops of each defense method.

[1] Li, Yuezun, et al. "Invisible backdoor attack with sample-specific triggers." (ICCV 2021)
[2] Zeng, Yi, et al. "Narcissus: A practical clean-label backdoor attack with limited information." (ACM CCS 2023)
[3] Wang, Tong, et al. "An invisible black-box backdoor attack through frequency domain." (ECCV 2022)

**Questions:**

1. Please refer to the weaknesses above.
 2. Some questions regarding experimental details:
- Why are different poisoning rates used for different attacks/datasets? I am not sure the comparison is fair given the varying settings.
- Why are the warping field parameters of WaNet strengthened?
- What are the backdoor features added to the backdoor probe set? Are they all the triggers evaluated in the experiments? If so, could the method really work with unseen triggers? (I might be confused here, because the method is claimed to can "zero-shot transfer to previously unseen backdoor attacks", but the paper does not really explicitly mention which backdoor features are used to record the loss trajectories and which are unseen ones.)

---

> ### Author Response · Authors · 2023-11-20
> **Addressing Weaknesses**
>
> Thank you for the review. We’re grateful for the feedback, and we’re pleased that you found our central contribution to be novel and interesting. Hopefully, we will be able to address your remaining reservations.
>
>
> > [method's presentation, major] I personally find the presentation of Section 3 quite hard to follow since there are no algorithm or figure to describe the method, or even a formulation.
>
> Thank you for the feedback. Following our top-level comment, we revised the paper and improved our presentation of our methods. Could you take a look and let us know if this is easier to understand?
>
>
> > [lack of experiments, major] There are only 3 types of backdoor attack (patch-based, blending-based, warping-based) that are considered in the experiments, so I am not sure if the defense is effective with all attacks. I think there should be more backdoor attack approaches included in the experiments as well as related works discussion, such as sample-specific ([1]), optimized trigger ([2]), or frequency domain attack ([3]). Moreover, there is no abaltion study/discussion about different choices for the hyperparameters used in the paper (detection threshold, k for the kNN classifier).
>
> Thank you very much for the references to the literature. We would like to note that we found the Sleeper Agent attack (another optimized-trigger attack and a predecessor to the Narcissus attack) did not generalize well to new training scenarios and architectures, and as a result did not evaluate it against the defenses when it failed to get a high ASR even without any defense.
>
> However, we agree with your overall criticism. While we considered many types of attacks, we wanted to keep the set of attacks relatively limited for this work, specifically focusing on ones that are easy to define and tune, as we focus on the multi-attack setting. We agree that a broader set of attacks would improve the robustness of our results.  We’d greatly appreciate any further suggestions about which ones would best complement our current results.
>
>
> > [underwhelming experimental results, major] The clean acc. of BaDLoss is significantly degraded in the cases of SIG, WaNet, and multi-attack on CIFAR10. With those underwhelming clean acc. (~60%), I doubt that the model can be considered functional, especially on such "easy" dataset like CIFAR10.
>
> We agree that the degraded clean accuracy is a major limitation of this work. However, we would like to note that the poisoning ratios of the Sigmoid and Warping attacks are higher than the other attacks. As a result, in order to remove the backdoor, many training examples must be removed – thus degrading the clean accuracy. Future work could, for instance, use self-supervised learning on removed examples so as to ensure that the information within those examples is not entirely lost, while supervised learning on only the clean examples. We plan to run additional experiments sweeping across poisoning ratios and detection thresholds – we expect that this will demonstrate superior performance for BaDLoss, on account of our better detection AUC.
>
>
> > [results' presentation, minor] There are many numbers in Table 2 and Table 3 but the best results are not highlighted. The authors should highlight the best results, or maybe report the average clean acc. and ASR drops of each defense method.
>
> Thank you for the feedback, we have updated the multi-attack table with average ASR and highlighted the best results in that column.

---

> ### Author Response · Authors · 2023-11-20
> **Answering Questions**
>
> Finally, thank you for the questions regarding our choices of attacks. We hope this clarifies the decisions we made in the selection process. Please let us know if you want us to provide any further points of information.
>
>
> > Why are different poisoning rates used for different attacks/datasets? I am not sure the comparison is fair given the varying settings.
>
> Different attacks require different poisoning ratios on different datasets. The Warping attack, if set at 1% poisoning ratio, is not learned (as the model simply memorizes the examples with the changed label, as opposed to learning the complex warping features needed to generalize to unseen warped attack examples). Similarly, while the Patch attack is fairly strong on CIFAR-10 (1% is more than enough for the attack to be learned, and even 0.1% would suffice under the right training conditions), it is much weaker on GTSRB (likely due to the higher image resolution – a 4-pixel patch is a proportionally greater feature in a 32x32=1024-pixel image than in a 224x224=50176-pixel image.) Each of these choices is motivated by wanting the attack to succeed on the target dataset, assuming no defense. Where possible, we used settings as close to their original papers for all attacks.
>
>
> > Why are the warping field parameters of WaNet strengthened?
>
> Similarly, because the warping field parameters used in the original paper led to a weaker attack (~50% attack success rate) in our experiments.
>
>
> > What are the backdoor features added to the backdoor probe set? Are they all the triggers evaluated in the experiments? If so, could the method really work with unseen triggers? (I might be confused here, because the method is claimed to can "zero-shot transfer to previously unseen backdoor attacks", but the paper does not really explicitly mention which backdoor features are used to record the loss trajectories and which are unseen ones.)
>
> As we noted, the backdoor feature added to the backdoor probe set is always a rotated version of the Patch attack in the multiattack setting. As this generalizes to very different backdoor attacks (e.g. the Warping attack), we make the claim that this allows BaDLoss to generalize onto new attacks. We also updated the methodology of our paper in order to make it easier to understand.

---

### Official Review · Reviewer_4YdZ · 2023-10-30

**Soundness:** 2 fair
**Presentation:** 2 fair
**Contribution:** 2 fair
**Rating:** 3
**Confidence:** 4

**Summary:**

This paper proposes to use the loss dynamics of a training sample to detect whether it is a backdoored sample. The idea is to construct two sets of contrast samples including clean samples and randomized-label samples. A kNN classifier was then trained on different types of training loss trajectories as the detector model. Under one-attack and multiple-attack evaluations, the proposed method shows promising results compared to existing defenses like NC, AC, SS, ABL et al.

**Strengths:**

1. The use of loss trajectory to detect backdoor samples is an interesting direction;

2. The proposed detector seems quite easy to train and works reasonably well against different types of attacks;

3. The proposed method is compared with a set of existing defenses NC, AC, SS, FA, ABL, et al.

**Weaknesses:**

1. In section 3.2, it is not clear how the loss trajectories were collected and how the detector was trained. The authors mentioned 500 bonafide clean examples, and then another 250 randomized-labeled samples, so how many samples were needed to extract the trajectories and train the detector? Also, it is not clear how the 250 backdoored probes were crafted, i.e., using what backdoor features? It has been shown that a stronger backdoor trigger can overwrite a relatively weaker backdoor trigger, so here the choice of the backdoor feature will become vital.

2. It is not clear how to tune the threshold to reject a training sample, as the poisoning rate should not be known to the defender in advance. This potentially makes the proposed defense fail either low poisoning rates or high poisoning rates. I.e., if the poisoning rate is 40%, how it is possible to remove all the poisoned samples by determining the threshold?

3. It is not clear how the loss trajectory is defined and how the proposed method can be adaptive to different types of attacks. The authors argued that backdoored samples can have either slow or fast training speed, yet it is not clear how the proposed detector can identify both or even more subtle cases.

4. The restus of existing defenses in tables 1 and 1 are stranger, where it shows ABL and other defenses fail the most case, which I believe it is not the case in their original papers.

5. The proposed method failed the Sinusoid attack in Table 2, which was not sufficiently explained.

6. The considered backdoor attacks was far less than in recent works [2,3].

7. The proposed method was not conompared with the SOTA backdoor sample detection method Cognitive Distillation [3], which can be applied to detect both training and test samples, yet the proposed method can only detect training samples.


[1] Wu, Dongxian, and Yisen Wang. "Adversarial neuron pruning purifies backdoored deep models."  NeurIPS, 2021.

[2] Li, Yige, et al. "Reconstructive Neuron Pruning for Backdoor Defense." ICML, 2023.

[3] Huang, Hanxun, et al. "Distilling Cognitive Backdoor Patterns within an Image." ICLR, 2023.

**Questions:**

1. How robust is the proposed defense to adaptive attacks, ie.., adversarially enhanced backdoor samples to evade the detector?

2. The authors mentioned MAP-D, but what is MAP-D was not clearly defined.

3. How to defend a high poisoning rate like 10% or even 20%?

4. Did the authors tune the baseline defenses on the tested attacks, the comparison was unfair if not.

5. How to choose a proper k for the knn detector?

6. which DNN model was used for CIGAR-10, whose clean ACC is too low.

7. A high-resolution dataset like an ImageNet subset should also be tested in the experiments, as they have different convergence speed and hen loss dynamics.

**Details Of Ethics Concerns:**

No theics concerns.

---

> ### Author Response · Authors · 2023-11-20
> **Addressing Weaknesses**
>
> Thank you for the review, and we’re glad that you found our core idea to be an interesting direction. We’re very grateful for your detailed feedback.
>
>
> > In section 3.2, it is not clear how the loss trajectories were collected and how the detector was trained. The authors mentioned 500 bonafide clean examples, and then another 250 randomized-labeled samples, so how many samples were needed to extract the trajectories and train the detector? Also, it is not clear how the 250 backdoored probes were crafted, i.e., using what backdoor features? It has been shown that a stronger backdoor trigger can overwrite a relatively weaker backdoor trigger, so here the choice of the backdoor feature will become vital.
>
> Thank you for the feedback. As we indicated in our top-level comment, we’ve improved our presentation of our methods. Would you please take a look at the revision and see if this more clearly communicates how our method functions?
>
>
> > It is not clear how to tune the threshold to reject a training sample, as the poisoning rate should not be known to the defender in advance. This potentially makes the proposed defense fail either low poisoning rates or high poisoning rates. I.e., if the poisoning rate is 40%, how it is possible to remove all the poisoned samples by determining the threshold?
>
> We agree that clearer instructions could be provided on how to select BaDLoss’ parameters. While we recommend certain pre-sets, these parameters could be tuned on the probe, e.g. using LOO classification and maximizing detection performance at a certain FPR/FNR. We have added a short discussion on how this might be done.
>
> Additionally, we note that most other methods have very weak threshold tuning tools, and BaDLoss’ adjustment is very flexible. We chose our aggressive threshold due to the problem of the unknown poisoning ratio. Low or high poisoning ratio attacks will often exhibit very anomalous trajectories (e.g. as exemplified by the warping attack, covered in our discussion section) – and appropriate choice of probes allows us to detect those attacks effectively.
>
>
> > It is not clear how the loss trajectory is defined and how the proposed method can be adaptive to different types of attacks. The authors argued that backdoored samples can have either slow or fast training speed, yet it is not clear how the proposed detector can identify both or even more subtle cases.
>
> We curated two probe sets, one which is easier to learn than benign training examples, and one which is harder to learn than benign training examples. Furthermore, we use a very aggressive filtering threshold of 0.1 on the output probabilities of our k-NN classifier. Therefore, this allows us to adapt to different types of attacks.
>
>
> > The results of existing defenses in tables 1 and 1 are stranger, where it shows ABL and other defenses fail the most case, which I believe it is not the case in their original papers.
>
> Regarding Table 1: We note that most of these defenses are not optimized to present a sensible AUC score. We generated AUC scores by sweeping over the relevant detection threshold to get false-positive/false-negative rates – but the detection threshold used by many of these methods is not robust to different settings if the practitioner wants to adjust the defense to have a higher FNR/FPR. This is not reporting that the defense cannot defend at all – indeed, ABL does defend against some attacks, especially on GTSRB – it is merely reporting that the defense is relatively fragile and cannot be adjusted easily to have stronger/weaker detection.
>
> For a longer explanation of why the ABL scores in particular are poor, please refer to our response to Reviewer QKR4 (final reviewer).
>
>
> > The proposed method failed the Sinusoid attack in Table 2, which was not sufficiently explained.
>
> Correct, and thank you for pointing this out. We’ve added a short explanation of why this is the case.
>
>
> > The considered backdoor attacks was far less than in recent works [2,3].
>
> This is true. However, we cover a range of different attack types. While more attacks would certainly make our results more robust, we are uncertain whether adding modern attacks would be particularly valuable. As we indicated in the paper, we tried to replicate the Sleeper Agent attack and failed to get reasonable ASR scores with the attack. We suspect that even relatively modern data poisoning attacks will be relatively fragile, and readily foiled by filtering-based defenses.
>
>
> > The proposed method was not conompared with the SOTA backdoor sample detection method Cognitive Distillation [3], which can be applied to detect both training and test samples, yet the proposed method can only detect training samples.
>
> Thank you for the feedback. While we will likely be unable to implement and run new SotA defense methods during the rebuttal period, we have added a section in the Appendix on how BaDLoss could be extended to detect backdoored test samples as well.

---

> ### Author Response · Authors · 2023-11-20
> **Answering Questions**
>
> Thank you for the questions. Please let us know if you have any further questions that we can help answer.
>
>
> > How robust is the proposed defense to adaptive attacks, ie.., adversarially enhanced backdoor samples to evade the detector?
>
> Adaptive attacks are not in our threat model – they are out of scope for this work. In our threat model, the attacker is only allowed to insert images into the training set, and they are unable to access any part of training or detection. We argue that this is a reasonable threat model, as the training process often happens internally in the same organization that eventually deploys the model (see Carlini et al. 2023 for an interesting example). Adversaries in such settings could far more easily modify data sources or storage than the training script.
>
> Carlini, N., Jagielski, M., Choquette-Choo, C.A., Paleka, D., Pearce, W., Anderson, H., Terzis, A., Thomas, K. and Tramèr, F., 2023. Poisoning web-scale training datasets is practical. arXiv preprint arXiv:2302.10149.
>
>
> > The authors mentioned MAP-D, but what is MAP-D was not clearly defined.
>
> We cite the paper (https://arxiv.org/pdf/2209.10015.pdf) in which it was introduced. We have revised the framing of the method, hopefully between the original paper and our improved presentation of our method, this is more clearly communicated. Apologies for any confusion.
>
>
> > How to defend a high poisoning rate like 10% or even 20%?
>
> High poisoning ratio attacks are very challenging to defend. Even a 99% successful filtering method can fail if 1% of a high-ratio attack slips through. We note that only BaDLoss reliably succeeds at the high-ratio attack in our evaluations: the warping attack.
>
>
> > Did the authors tune the baseline defenses on the tested attacks, the comparison was unfair if not.
>
> We adjusted the baseline defenses as described to improve their performance in our chosen setting, but we used the detection thresholds provided in their respective papers when not specified. This is our understanding for how a practitioner would implement these defenses. As a result, we do not think this comparison is unfair.
>
>
> > How to choose a proper k for the knn detector?
>
> We suggest that k be chosen to maximize performance of the detector on the probe examples (e.g. through LOO classification). We have added a comment to this effect.
>
>
> > which DNN model was used for CIFAR-10, whose clean ACC is too low.
>
> We use a ResNet-50. We train without augmentations throughout, likely contributing to a relatively lower clean accuracy. We’ve added a comment to this effect in the paper.
>
>
> > A high-resolution dataset like an ImageNet subset should also be tested in the experiments, as they have different convergence speed and hence loss dynamics.
>
> We agree that training on a large-scale high-resolution dataset is interesting. We cover the high-resolution aspect using GTSRB, where we upscaled all GTSRB images to 224x224. Although it is easy for us to report the performance of our method on ImageNet, transferring all baselines to this larger dataset is non-trivial as this will require hyperparameter tuning for each of these individual baselines.

---

> ### Comment · Reviewer_4YdZ · 2023-11-22
> **Thanks for the response**
>
> I would like to thank the authors for the rebuttal. However, my concerns remain regarding the following 3 points:
>
> 1. Missing a clear definition, thorough understanding, and adaptive evaluation of the loss dynamics as a detection metric.
>
> 2. Weak evaluation against a very limited number of backdoor attacks.
>
> 3. Lack of comprehensive comparison to current SOTA methods.
>
> I thus would like to keep my initial rating.

---

### Official Review · Reviewer_U6eM · 2023-10-31

**Soundness:** 2 fair
**Presentation:** 2 fair
**Contribution:** 2 fair
**Rating:** 3
**Confidence:** 3

**Summary:**

The paper proposes a backdoor detection method named BadLoss. It focuses on the threat model that modifies the training dataset and detect them via loss dynamics. Specifically, it needs a probe set which has potential trigger patterns and use them to detect poisoned samples. After deleting the backdoored samples, it uses the clean set to retrain the model. The experiments validate the effectiveness of their method.

**Strengths:**

1.	Addressing the datasets backdoor attack is still an interesting and realistic direction.
2.	Detecting multi-trigger backdoor attacks is an efficient way to deal with large scale dataset.

**Weaknesses:**

1.	Results have only marginal improvement. For example, the badloss cannot maintain a consistent high clean accuracy for the sinusoid attack.
2.	Ablation study is needed. For example, how do you choose the threshold and why you choose that. Would the probes set affect the performance significantly? How do you choose the k in kNN classifier.
3.	Using loss is not a novel idea to detect backdoors, and there are many similar works.
Li, Yige, et al. "Anti-backdoor learning: Training clean models on poisoned data." Advances in Neural Information Processing Systems 34 (2021): 14900-14912.
Guan, Jiyang, et al. "Few-shot backdoor defense using shapley estimation." Proceedings of the IEEE/CVF Conference on Computer Vision and Pattern Recognition. 2022.

**Questions:**

Please refer to the weakness part.

---

> ### Author Response · Authors · 2023-11-20
>
> Thank you for the review. We’re grateful for your detailed feedback.
>
> > Results have only marginal improvement. For example, the badloss cannot maintain a consistent high clean accuracy for the sinusoid attack.
>
> We agree, in general, with your comment on marginal improvements. We note that the warping and sinusoid attacks are particularly challenging to defend against consistently – and no defense consistently exceeds our performance. Addressing the sinusoid attack in particular, we note that BaDLoss defends against the attack adequately in CIFAR-10. However, the attack is far stronger in GTSRB, and is only defended against by the Frequency Analysis defense – and we comment on the fact that despite the fact that BaDLoss achieves highly precise detection, it is still hard to defend against the attack. However, we highlight that, in this case, the Frequency Analysis defense shows a much higher clean accuracy degradation than BaDLoss in order to achieve that defense. We maintain that BaDLoss is at least on the Pareto frontier of available defenses on the sinusoid attack.
>
> > Ablation study is needed. For example, how do you choose the threshold and why you choose that. [...] How do you choose the k in kNN classifier.
>
> The filtering threshold as well as the value of `k` for k-NN are both key hyperparameters in our approach. We use a strong filtering threshold of 0.1 as it is sufficient for only a fraction of the images to pass through the filter for the backdoor to be learned. We experimented with other values, but found them to be less effective. The value of `k` in our case is mainly taken from prior work of Siddiqui et al. 2022, i.e., k=20. We experimented with minor variations, but found this to be a reasonable starting point. We agree with the reviewer that doing a principled evaluation of the impact of hyperparameters might yield useful insights for the reader.
>
> > Would the probes set affect the performance significantly?
>
> We are slightly confused by what the reviewer precisely mean here. It would be helpful to clarify if we understood it in the wrong way. Our method certainly depends on both the size as well as the fraction of the probes present in the dataset as we are leveraging training dynamics of the model. We chose the current sizes to ensure we can simulate both easier to learn as well as harder to learn patterns as compared to regular examples in the training set. After tuning the sizes of the probe sets, one needs to retune the other two main hyperparameters in our setup. However, given a good coverage of both easier as well as harder to learn patterns, we assume that our setup would be agnostic to the actual number of test backdoor instances, as far as they don’t start simulating trajectories of benign training examples.
>
> > Using loss is not a novel idea to detect backdoors, and there are many similar works. Li, Yige, et al. "Anti-backdoor learning: Training clean models on poisoned data." Advances in Neural Information Processing Systems 34 (2021): 14900-14912. Guan, Jiyang, et al. "Few-shot backdoor defense using shapley estimation." Proceedings of the IEEE/CVF Conference on Computer Vision and Pattern Recognition. 2022.
>
> Thank you for this feedback, and for the valuable references to other methods in the literature. We would like to put forward one important clarification: we treat the entire loss trajectory as meaningful, leading to much stronger identification. Methods like Anti-Backdoor learning use a point estimate of loss – which is considerably harder to draw meaningful conclusions from, as ABL’s mixed performance in our evaluations demonstrates.
>
> Additionally, we would like to note that the Shapely Value Estimation method for backdoor defense does not actually directly use the model’s loss values to identify backdoored examples – instead, it uses trigger-reversing similar to Neural Cleanse to identify the model’s backdoors, then use their Shapely Value Estimation method to identify the neurons that most contribute to that reversed backdoor, as expressed via attack success rate. While the idea that backdoored examples might have lower loss is implied (through the trigger-reversing assumption that backdoored examples get low loss values on their corresponding class), we do not believe it is meaningfully explored in this work.

---

> > ### Comment · Reviewer_U6eM · 2023-11-22
> > **Thanks for the response**
> >
> > Thanks the authors for the rebuttal. However, the response cannot convince me of the effectiveness with respect to experiments:
> > 1. Ablation study is missing. It is hard to tell if the model is sensitive to hyperparameters.
> > 2. The improvement is marginal.
> >
> > I thus would like to keep my rating unchanged.

---

### Author Response · Authors · 2023-11-20
**Improved Methods Presentation**

Hello all,

Thank you everyone for the detailed reviews. We’d like to address one change in particular that we are making to all reviewers.

One common piece of feedback was that the presentation of our method was unclear. We’ve created a new figure that visualizes the BaDLoss algorithm end-to-end, and added a more detailed description of the algorithm used in the Appendix. We hope this clarifies our methodology.

---

### Meta-Review · Area_Chair_jqCv · 2023-12-08

**Metareview:**

The paper presents an innovative idea in backdoor detection, leveraging loss dynamics. However, the reviewers consistently raised issues about the paper's practicality, the breadth of its experimental evaluations, and its effectiveness against a range of backdoor attacks. There are also concerns about the method's presentation and the need for more comprehensive comparisons with current state-of-the-art methods. The reviewers suggest that the paper requires significant improvements in these areas before it can be considered for acceptance.

**Justification For Why Not Higher Score:**

As said in the meta review

**Justification For Why Not Lower Score:**

NA

---

### Decision · Program_Chairs · 2024-01-16

Reject